# Preparation of Ionic Liquid-Coated Graphene Nanosheets/PTFE Nanocomposite for Stretchable, Flexible Conductor via a Pre-Stretch Processing

**DOI:** 10.3390/nano10010040

**Published:** 2019-12-23

**Authors:** Yu Zhang, Kaichang Kou, Tiezheng Ji, Zhengyong Huang, Shuangcun Zhang, Shijie Zhang, Guanglei Wu

**Affiliations:** 1Ministry of Education and Shaanxi Key Laboratory of Macromolecular Science and Technology, School of Science, Northwestern Polytechnical University, Xi’an 710129, China; yuz@mail.nwpu.edu.cn (Y.Z.); koukc@nwpu.edu.cn (K.K.); tzji@nwpu.edu.cn (T.J.); zhangsc@nwpu.edu.cn (S.Z.);; 2State Key Laboratory of Power Transmission Equipment & System Security and New Technology, Chongqing University, Chongqing 400040, China; 3Institute of Materials for Energy and Environment, State Key Laboratory of Bio-Fibers and Eco-Textiles, College of Materials Science and Engineering, Qingdao University, Qingdao 266071, China; 4Key Laboratory of Engineering Dielectrics and Its Application, Ministry of Education, Harbin University of Science and Technology, Harbin 150080, China

**Keywords:** PTFE nanocomposites, graphene, ionic liquid, pre-stretching, tensile properties, electrical conductivity, thermal conductivity

## Abstract

The various volume concentrations of ionic liquid-modified graphene nanosheets filled polytetrafluoroethylene nanocomposites (IL-GNs/PTFE) for flexible conductors were fabricated via a pre-stretch processing method after cold-press sintering. The results indicated that pre-stretching has no significant weakening in the electrical conductivity of the nanocomposites, while the Young’s modulus greatly reduced by 62.5%, which is more suitable for flexible conductors. This may be because the reduced conductivity by the destructive conductive pathway cancels out the enhanced conductivity by the increased interlamellar spacing of IL-GNs via a pre-stretch processing, and the nanocomposite exhibits a phase transition from two to three-phase (with the introduction of an air phase) during pre-stretching. It was also found that the tensile strength of the nanocomposites was enhanced by 42.9% and the elongation at break and thermal conductivity decreased slightly with the same filler content after pre-stretching. The electrical conductivity of the pre-stretched nanocomposites tended to stabilize at 5.5 × 10^−2^ s·m^−1^, when the volume content of the packings achieved a percolation threshold (1.49 vol%). Meanwhile, the electrical resistivity of the pre-stretched 3.0 vol% IL-GNs/PTFE nanocomposite was slightly reduced by 0.30%, 0.38%, and 0.87% respectively after 180° twisting, 180° bending, and 10% stretching strain for 1000 cycles.

## 1. Introduction

Nowadays, the flexible and stretchable conductor, which is a fastest developing material, is one of the most widely used in the electronics industry, such as flexible sensors, stretchable supercapacitors, health monitoring installation, and wearable devices [1,2,3,4,5]. Generally, polymers are regarded as excellent insulating and soft materials in comparison with metal. Hence, if appropriate conductive fillers (such as metal powder [6,7], carbon black [8,9,10], carbon nanotubes [7,11,12]) cooperate with the polymeric matrix and the contents of the filler reach the percolation threshold, which means that the fillers in contact with each other in the matrix form a conductive network, it completes the transition from insulating material to semiconductor or conductor material.

Consequently, with the objective of actually fabricating high-performance flexible conductors, respectable researchers have been devoted to exploiting polymer-based conductors with flexibility and stretchability [13,14,15,16,17]. For instance, Liu and co-workers [18] transferred silver nanowires (AgNWs) into polydimethylsiloxane (PDMS) via an effective method to obtain AgNW/PDMS stretchable electrodes of maintained low sheet resistance after stretching, twisting, and bending for 1000 cycles. Hwang et al. [19] fabricated sandwich structure films of AgNWs hybridized with 0.025 wt% single-walled carbon nanotubes (SWCNTs) between layers of PDMS and transparent polyurethane (TPU), with the observed low sheet resistance value of 30 Ω/sq.

In recent years, graphene nanosheets (GNs) have attracted more attention as a hopeful packing to fill polymeric matrixes for electrically conductive properties, due to their unique electrical conductivity [20,21,22]. Yan et al. [23] designed and prepared flexible conductive bilayer films composed of elastomeric copolymer (*N*-isopropylacrylamide and 2-methoxyethyl acrylate) and wrinkled graphenes, which exhibited a high electrical conductivity of 126 s·cm^−1^. Yang et al. [24] fabricated a novel polyimide-based graphene foam via chemical and thermal reduction after dip-coating a polyimide foam to achieve 0.4 s·m^−1^ in electrical conductivity. However, the most critical challenge in the fabrication is the uniform dispersion of graphene nanosheets in the polymer matrix, which have a poor dispersion in solvent due to agglomeration [25,26,27]. In order to improve the dispersibility of GNs in substrate, significant scholars prepared graphene into reduced graphene oxide (rGO) [28,29] or grafted the hydroxyl group (–OH) [30], epoxy group (–CH(O)CH–) [31], carboxyl group (–COOH) [31,32], and other functional groups at the edge or defect of graphite oxide. Although these methods can prepare covalently bonded graphene with stable dispersion ability, the oxygen-containing functional groups abate the electrical performance of graphene. However, it has been reported that the functionalization of non-covalent bonds can be used to prepare graphene with stable dispersion and a less damaged lattice structure while retaining its excellent properties [33,34]. As a solvent with a small molecular volume, ionic liquid (IL) is often used as a non-covalently functionalized modifier, which can be better inserted or adsorbed into the nanopackings and can effectively prevent the agglomeration of the filler [35,36]. Zhao and co-authors had reported that the graphene sheets were functionalized with a synthetic poly(1-vinylimidazole) type ionic liquid by a quaternarization reaction, which showed improved dispersibility in DMF (Dimethyl Formamide) and ethanol solution [37].

As is known to all, polytetrafluoroethylene (PTFE) has unique properties due to its high fluorine–carbon bond energy, which has attracted extensive attention in the industrial field with its required thermostable performance [38]. Nevertheless, up to now, most of the investigations on PTFE have focused on improving its disadvantages, such as poor creep resistance, while the few references on the electrical conductivity of PTFE composites are available. In addition, due to the very regular molecular chains of PTFE and their high crystallinity after sintering, the PTFE matrix composite has a relatively high hardness that could not be suitable as a flexible material. However, through our previous study, it was found that when the PTFE-based composites were stretched within a specific range for a fast speed at ambient or high temperature, the characteristics of the material would change from rigidity to flexibility [39,40]. Hence, our purpose of this experiment is the preparation of a flexible conductor. First, we need to evenly disperse graphene nanosheets in the acetone; then, we need to prepare the PTFE nanocomposite by using the traditional cold-pressing and sintering process after mixing the ionic liquid-modified graphene nanosheets (IL-GNs) with PTFE powders by the wet-mixing method in the acetone. Finally, we need to soften the nanocomposite by a pre-stretching treatment, which can make it more suitable as a conductor nanocomposite (IL-GNs/PTFE) with flexibility and stretchability. Meanwhile, the influence of pre-stretching on the tensile, thermal, and electrical properties of the IL-GNs/PTFE nanocomposite and the conductivity of flexible conductors after twisting, bending, and stretching were investigated.

## 2. Experimental Procedure

### 2.1. Preparation of IL-GNs

Since we prepared the nanocomposite by the wet mixing method, we first needed to prepare GNs that could be dispersed uniformly in acetone solution. The specific preparation methods are as follows. Firstly, the commercial expansible graphite (75 μm, Shanghai Aladdin Biochemical Technology Co., Ltd., Shanghai, China) was transferred into a crucible and heated for 60 s in a preheated furnace at 800 °C. Subsequently, the obtained expanded graphite (EG) was mixed with 1-butyl-3-methylimidazolium hexafluorophosphate (BMIMPF_6_, 97%, Shanghai Macklin Biochemical Technology Co., Ltd., Shanghai, China) at a mass ratio of 1:1, and then added into ethanol solution (Xi’an Chemical Reagent Corporation, Xi’an, China) for ultrasonic dispersion for 1 h. Finally, the mixed solution was added into the jar mill with mechanical exfoliation for an hour before being dried at 70 °C for 2 h to constant weight. The fabrication of graphene nanosheets modified with ionic liquid (IL-GNs) is described in Scheme 1.

### 2.2. Fabrication of IL-GNs/PTFE Nanocomposites for Flexible Conductors

Firstly, the IL-GNs/PTFE nanocomposite specimens were prepared by wet mixing in acetone solution and then cold pressing and the sintering method. Secondly, the pre-stretching method was used to fabricate the flexible conductors, which is consistent with the PTFE composite preparation method in our previous literature [40,41]. In these experiments, the IL-GNs/PTFE nanocomposites with IL-GNs volume fractions of 0.5%, 1%, 1.5%, 2%, 2.5%, and 3% (the graphite density 2.2 g·cm^−3^ can be used to convert a weight fraction into a volume fraction.) for flexible conductors were fabricated via high-temperature (180 °C) unidirectional pre-stretching treatment to 1.5 fold of the original length at a rate of 100 mm·min^−1^ and remained stretched for 10 min before quenching. The unstretched test bar was 50 mm in length, 10 mm in width, and 2.2 mm in thickness, and it can be transmuted into the strip-shape sample with approximate dimensions of 55 mm length, 10 mm width, and 2 mm thickness after pre-stretching treatment due to the shrinkage of the polymer molecular chains. The fabrication of the IL-GNs/PTFE flexible conductor nanocomposite is also shown in Scheme 1.

### 2.3. Characterization Methods

Fourier transform infrared (FT-IR) spectra were recorded by a RAYLEIGH WQF-510A Fourier transform infrared spectrometer (Beijing, China) in the range from 4000 cm^−1^ to 400 cm^−1^. X-ray diffraction (XRD) was carried out using an X’Pert PRO MPD X-ray diffraction spectrometer (Almelo, The Netherlands) in the interval 2θ between 10° and 60° with Cu Kα radiation. Transmission electron microscopy (TEM) and scanning electron microscopy (SEM) images were made using a FEI Talos F200X TEM (Hillsboro, OR, USA) and FEI Verios G4 (Hillsboro, OR, USA), respectively.

The tensile properties of nanocomposites were investigated following ASTM D638-14 in the SANS CMT7204 tensile test machine (Shenzheng, China). A high precision digital scleroscope WHW MC010 (Shanghai, China) was used to test for shore hardness at a dead load of 5 kg and a residence time of 15 s according to ASTM D2240-2005. The density (*ρ*) of nanocomposites was characterized by the Archimedes method according to ASTM D792-2008. Thermal conductivity was measured using a thermal conductivity testing instrument Netzsch LFA427 (Selb, Germany) by the flash method according to standard test method ASTM E1461-2013. The specimen geometry is a test cube with a length of 10 mm and a thickness of 2 mm. The crystallization behavior of nanocomposites was employed by a TA Q600ADT (New Castle, DE, USA) different scanning calorimetry (DSC) instrument, and the heating rate was 10 °C·min^−1^. The calculation method of crystallinity adopted the equation recorded in the previous article. Six pieces of data were collected for these tests, and the reliability of the data was ensured by calculating the average.

The electrical resistivity of IL-GNs/PTFE nanocomposites was measured on a four-point conductivity probe with a linearly arranged four-point head. The volume resistance of nanocomposites in various temperatures was recorded by a self-made RT109A high-precision resistivity tester (non-continuous recording equipment).

## 3. Results and Discussion

### 3.1. Characterization Analysis of IL-GNs

Figure 1 shows the absorption peaks of BMIMPF_6_, GNs, and GNs surface-modified by BMIMPF_6_ (IL-GNs) in the FT-IR spectra, respectively. The characteristic peaks of both BMIMPF_6_ and IL-GNs curves are observed at 3172, 1385, 1169, 748, and 555 cm^−1^, corresponding to the vibration of methine (C–H) in cyclic BMIM^+^. The bands at 842 and 1574 cm^−1^ are attributed to the stretching vibration of PF_6_^−^ and the carbon–nitrogen bond (C–N) inside BMIMPF_6_, respectively [42,43,44,45]. In addition, the FT-IR spectrum of GNs shows the bands at 1649 cm^−1^ for carbonyl (C=O) [46], which means that GNs were partially oxidized to graphene oxide (GO) during the heat treatment of expandable graphite. These results clearly indicate that ionic liquids have been successfully inserted into GNs.

Figure 2 shows the XRD pattern of expanded graphite (EG), GNs, and GNs surface-modified by ionic liquids (BMIMPF_6_) (IL-GNs). XRD is used to investigate the effect of surface treatment on the plane of GNs. By comparing the XRD curves of the three, it is found that there is a diffraction peak at 2θ = 26.5°, which correspond to the characteristic plane (003) of graphene [47,48]. However, the diffraction peak strength of EG is significantly smaller than that of the other two, which may be because the excessive layer spacing affects the test results of XRD. Furthermore, the diffraction peak of IL-GNs was slightly leftwards compared with that without ionic liquids. This consequence has further confirmed that ionic liquids facilitate a complete mechanical exfoliation.

The interlamellar spacing (*d*) between GNs can be calculated by using the Bragg equation [49,50]:(1)nλ=2dsinθ
where *n* represents the diffraction order (usually, *n* = 1), *λ* = wavelength of the X-rays (0.154056 nm), and *θ* = diffraction angle. The calculation results are represented in Table 1. It can be seen from the calculation results that the *d* of IL-GNs is smaller than that of expanded graphite and GNs, indicating that mechanical exfoliation and ionic liquid have a good effect on improving the agglomeration of GNs.

Through SEM technology, it can be noted from Figure 3a that the high-temperature expansion converted the expansible graphite into a worm-like expanded graphite consisting of loosely jointed thin GNs. Figure 3(b1,c1) shows that the worm-like structure completely disappeared after mechanical exfoliation, and the graphene nanosheets came to exist in a single or multi-layer crimped morphology. The SEM image of GNs (Figure 3(c1)) with horizontal dimensions over 10 μm interconnected edges by edges via van der Waals forces after mechanical exfoliation is quite different from that of IL-GNs (Figure 3(b1)). The TEM (Figure 3(b2,c2)) photographs also show that the delamination effect of the IL-GNs was better than that of the GNs.

Moreover, the sample photograph in Scheme 1a shows the dispersity of GNs and IL-GNs in acetone solution after 24 h. Apparently, the GNs modified by BMIMPF_6_ formed a uniformly and stably distributed black suspension in acetone, while the GNs were deposited at the bottom of the suspension due to agglomeration. This may be because BMIMPF_6_ functionalized GNs via non-covalent π–π interactions between the imidazole ring and basal plane of GNs, which leads to the interlamination of GNs that existed in coulomb forces and repelled each other. Scheme 1b shows that the IL-GNs have a better dispersion in PTFE powders than that of GNs after the wet-mixing method.

### 3.2. Tensile Properties of PTFE-Based Nanocomposites

The strain–stress curves of the PTFE-based nanocomposites with different packing volume contents are presented in Figure 4, and the corresponding tensile properties are listed in Table 2. It is obvious that the pre-stretch processing has a remarkable influence on the tensile properties. In particular, it is worth noting that the Young’s modulus of unstretched nanocomposites decreased sharply from approximately 48 MPa to about 18 MPa of pre-stretched ones, which has largely reduced by 62.5%. Generally, PTFE with higher modulus and hardness has a larger area of crystallization. However, in our experiment, the crystallinity of the pre-stretched nanocomposites was nothing breathtakingly different from that of the unstretched ones (As shown in Table 3, according to the results of crystallinity calculation, the crystallinity of the pre-stretched nanocomposite basically agreed with that of the unstretched one, so it can be judged that the pre-stretched treatment has not destroyed the crystal region of the material). Therefore, it was speculated that the large crystal region in the PTFE substrate would be split into multiple microcrystalline regions due to the pre-tensile treatment, which may be an important factor in the softening of nanocomposites. In addition, the spiral structure of the PTFE molecular chain in the high crystallinity region can be seen as a “miniature spring” with a locked deformation ability, whose freedom has been provided by the fragmented crystalline regions. Combined with the above two possible reasons, the PTFE-based nanocomposite presents a transition from rigidity to flexibility. Pre-stretching can also transparently enhance the tensile strength of the nanocomposites from approximately 14 MPa to about 20 MPa (increase by 42.9%), which contributes to the orientation of PTFE molecular chains, while the elongation at break declined from approximately 340% to 310% (decreased by 8.8%), which was due to the increased defects (microvoids) resulting from pre-stretching.

In addition, the maximum additive amount of filler only accounts for 3 vol% of the nanocomposite, and the low addition of filler does not affect the continuity of the PTFE matrix molecular chain. Hence, the packings have little effect on the tensile properties of nanocomposites, which is far less than the effect of pre-stretching.

The shore hardness and density of the PTFE-based nanocomposites with different packing volume contents are listed in Table 3. It was detected that the hardness and density of the pre-stretched nanocomposites are lower than those of the unstretched ones, which was consistent with a previous conclusion of Young’s modulus variation. Furthermore, from the SEM of the nanocomposite fracture surface (Figure 5), it can be clearly observed that the microvoids were induced by pre-stretch processing in the polymer matrix, which may be another key factor for the softening of the nanocomposite. These micrographs further explain that the “freedom-space” (microvoids) for the “miniature spring” are formed during the pre-stretch processing.

### 3.3. Thermal and Electrical Conductivities of PTFE-Based Nanocomposites

Figure 6 presents the thermal conductivities (λ) of the PTFE-based nanocomposites. It can be noted that the thermal conductivity of all the nanocomposites increases significantly with the increase of the packing content. This variation is consistent with the expectation: the packings with higher thermal conductivity can significantly improve the thermal conductivity of the nanocomposites, and the higher the content density of the packings, the higher the thermal conductivity of the composite. However, it was also found that the λ decreased slightly after pre-stretching. For instance, we could see the λ for the 1.5 vol% pre-stretched nanocomposite achieved 0.428 W·m^−1^·K^−1^, which was lower than that of the unstretched one (0.440 W·m^−1^·K^−1^) with the same packing volume content. This variation can be ascribed to two reasons: On the one hand, as the contact area of the thermal conductive channels formed by the packings decrease or disconnect due to the pre-stretching treatment, the heat is more likely to accumulate in the PTFE substrate with lower thermal conductivity. The variation of thermal pathways could be observed in Figure 7. It is easy to detect from Figure 7a that the packed IL-GNs in the unstretched nanocomposite are tightly bound together, while the contact area of packings is reduced or directly interrupted after pre-stretching, as shown in Figure 7b. On the other side, the two-phase (packings and polymer matrix) nanocomposite is transformed into a three-phase one during the pre-stretching; that is, the air phase (microvoids) with poor thermal conductivity is introduced into the initial two phases.

Generally, a conductive polymer is prepared by adding conductor filler, and the destruction of the conductive path in a polymer matrix would seriously affect the conductive property of the composite. However, the results of this experiment show an interesting phenomenon. As can be seen from Table 4, the electrical conductivities of pre-stretched nanocomposites are almost the same as that of the unstretched one, which means that the pre-stretch treatment did not have a serious impact on the electrical conductivity of the nanocomposite. These experimental results contradict the theory that the reduced conductivity was caused by the specimen irreversibly lengthening during pre-stretch processing.

Combined with XRD analysis (Figure 8 and Table 4), it was found that the interlamellar spacing (*d*) of IL-GNs in the unstretched matrix was smaller than that after pre-stretching. This may be because ionic liquid intercalated graphene has been pressed tightly by the cold-pressing during sample preparation (The *d* of IL-GNs in unstretched nanocomposites is less than that of IL-GNs powders, as shown in Table 1). Adding graphene thickness would inevitably increase the probability of the transferred electrons being exposed to the defects, which consists of imperfect graphene nanosheets. Zhang et al. [51] has reported that increasing the interlayer spacing of graphene could enhance the electrical conductivity of composites. Hence, the pre-stretching treatment would have a negative impact on the conductive pathways as a matter of course, but at the same time that the increase in the packing interlamellar spacing would have a positive impact on the enhancement of electrical conductivity, which may cancel each other out. As a result, the pre-stretching would not seriously affect the electrical conductivity of the IL-GNs/PTFE nanocomposites.

### 3.4. Percolation Threshold of Pre-Stretched Nanocomposites

Figure 9 represents the variation in the electrical conductivity of the pre-stretched IL-GNs/PTFE nanocomposites as a function of the IL-GNs concentration. In order to ensure the reliability of data, the electrical conductivity measured values were performed on at least three specimens and nine test points for each IL-GN content. Similar to the curves of typical carbon packed polymer matrix composites, the electrical conductivity increases sharply with the increase of IL-GNs content at a certain volume fraction due to the conductive filler forming a definite interconnection paths in the insulating substrate, which is the percolation threshold ($\phi_{c}$). It is evident from Figure 9 that when the addition of graphene is in excess of 1.0 vol%, the electrical conductivity of the composite increases rapidly, and the electrical conductivity increased by nine orders of magnitude as the IL-GNs concentration increased from 1.0 vol% to 1.5 vol%, which means that the formation of the interconnection paths within this scope could be identified. Meanwhile, the minuscule error bar for each specimen also indicates that the IL-GNs have been uniformly distributed in the PTFE matrix. A classical percolation model is used to correlate the percolation threshold ($\phi_{c}$), as indicated in Equation (2):(2)$$\sigma_{c}=\sigma_{f}{\left(\phi -\phi_{c}\right)}^{t}$$
where *σ_c_* is the electrical conductivity of the composite, *σ_f_* is the filler conductivity (graphene, 104.92 ± 0.52 ≈ 105 s·m^−1^ [52]), ϕ is the filler volume fraction, and *t* represents the exponential coefficient. The values of *t* = 1.99 and $\phi_{c}$ = 1.49 vol% could be obtained by fitting the test data with the above equation.

### 3.5. Electrical Conductivity under Cyclic Deformation

Deformability is a significant feature of flexible composites. In this work, twisting, bending, and stretching experiments were carried out on four kinds of IL-GNs/PTFE nanocomposites, which had a greater filler volume fraction than the percolation threshold, i.e., 1.5, 2.0, 2.5 and 3.0 vol%, and the electrical conductivity of the nanocomposites after deformation (twisting, bending, or stretching) 1, 100, 500, and 1000 times was measured respectively (The unstretched nanocomposites were not discussed due to the insufficient deformation capacity). Here, *R*_0_ represents the original resistance of the nanocomposite, and Δ*R*/*R*_0_ refers to the resistance change as a function of the deformation cyclic number. As shown in Figure 10a,b, the Δ*R*/*R*_0_ increases after twisting or bending several times respectively, and the increase in Δ*R*/*R*_0_ of a low IL-GNs volume content nanocomposite is greater than those of the material with high volume content. This may be due to the poor creep resistance of PTFE. It is all known that the molecular chains of PTFE are prone to slip under long-term pressure, resulting in the disconnection of the conductive path inside the composite material. Commonly, the addition of filler can prevent molecular chains slippage of the substrate to some extent, and the resistance stabilities of nanocomposites with high IL-GNs volume fractions are stronger than those with low volume fractions. Moreover, it should be also noted that the Δ*R*/*R*_0_ remains stable when the deformation increased from 90° to 180°, whether twisting or bending.

However, it is observed that the increase amplitude of Δ*R*/*R*_0_ is inappreciable even after 1000 twisting or bending releasing cycles. For instance, the Δ*R*/*R*_0_ was increased by 0.13% and 0.14% in the first 180° twisting and 180° bending, respectively, and it only further increased to 0.29% and 0.38% after 180° twisting and 180° bending for 1000 cycles, respectively. The explanation factor of this phenomenon could be that the pre-stretching treatment has better released the flexible characteristics of the PTFE molecular chain, and the deformation of the material in the later stage (twisting and bending) is not enough to have a decisive influence on the conductive path of the pre-stretched IL-GNs/PTFE nanocomposite.

So far, as we know, the resistivity of the flexible conductor increases rapidly with the increase of tensile strain, even less than 10% for the tensile ratio [53,54]. Therefore, the fabrication of a flexible conductive polymer with high tensile strain has become a research hotspot. In this study, we also conducted 1, 100, 500, and 1000 stretching–releasing cycles on the electrical conductivity of the IL-GNs/PTFE nanocomposites at the tensile ratios of 5% and 10%, respectively. Similar to the results of twisting and bending, the Δ*R*/*R*_0_ increases with the number of stretching cycles, and the Δ*R*/*R*_0_ for the 10% tensile ratio is significantly higher than that for 5% tensile ratio at the same IL-GNs volume content, as shown in Figure 10c. Interestingly, a significant increase in Δ*R*/*R*_0_ was noted as the number of stretching process cycles increased from 1 to 100. This may be due to the pre-stretching process carried out in the early preparation of nanocomposites, which resulted in the stretched PTFE matrix chains that have a good cushioning effect against the deformation in the direction of stretching in a short time.

### 3.6. Resistance Change in Stretching at Different Temperatures

Scheme 2 illustrates the custom-made test equipment. The volume resistivity could be calculated according to Equation (3) [55,56,57]:(3)$$R=\frac{US}{Ih}$$
where *R* is the volume resistivity, *U* is the voltage applied to the sample, *I* is the current passing through the sample, and *S* and *h* respectively represent the area of the sample in contact with the sensor and thickness of the specimen.

Figure 11a–c shows the Δ*R*/*R*_0_ of the three pre-stretched IL-GNs/PTFE nanocomposites with the highest electrical conductivity and materials after 100 stretching cycles at different temperatures as a function of the stretching strain. Apparently, the high volume fraction of IL-GNs has a slight advantage in deformation resistant capability compared to that of the low volume fraction. For example, the Δ*R*/*R*_0_ of the pre-stretched IL-GNs/PTFE nanocomposite decreases from 4.96% for 2.0 vol% IL-GNs to 3.43% for 3.0 vol% IL-GNs with the condition of 10% stretching strain at 25 °C (Similar changes occurred at 75 °C and 125 °C). The decreased Δ*R*/*R*_0_ should directly relate to the more densely conductive pathways, which is also evident from the tests of twisting and bending shown in Figure 10. Meanwhile, the Δ*R*/*R*_0_ of the nanocomposite is slightly increased after 100 stretching cycles under different temperatures (as shown by the solid line in Figure 11a–c). As can be noticed from Figure 11d, the high temperature has a significant effect on the Δ*R*/*R*_0_ of nanocomposites, the Δ*R*/*R*_0_ at low temperature has a lower value than that at high temperature with the same packing content normally due to the positive temperature coefficient (PTC). Compared to metal materials, non-metallic materials usually have a larger thermal expansion coefficient (CTE), resulting in the fracture of conductive paths and an increase of resistivity due to the wide difference in CTE between fillers (−8 × 10^−6^ K^−1^ for graphene [58]) and the polymer matrix (approximate 100 × 10^−6^ K^−1^ for PTFE [59]). It is interesting to observe that the growth rate of Δ*R*/*R*_0_ with low packing content is slightly higher than that of Δ*R*/*R*_0_ with high packing content, as shown by the dotted line in Figure 11d. While at 10% stretching strain, the ΔR/R_0_ increased from 4.95% to 8.27% (up 67.07%) at 2.0% filler volume fraction; in comparison, the Δ*R*/*R*_0_ increased from 3.43% to 5.72% (up 65.80%) at 3.0% filler volume fraction. Although the increase of packing content would reduce the Δ*R*/*R*_0_, this may be attributed to the nanocomposite with a higher content of high thermal conductivity fillers (IL-GNs) having better heat dissipation performance, which means that a high packing content nanocomposite with lower CTE can do better at preventing the destruction of conductive pathways in the matrix.

## 4. Conclusions

The IL-GNs were prepared by micromechanically exfoliating the expanded graphite (EG) in ionic liquid BMIMPF_6_. Then, PTFE matrix nanocomposites containing the IL-GNs with volume concentrations of 0.5%, 1.0%, 1.5%, 2.0%, 2.5%, and 3%, respectively, were fabricated via the cold-pressing and sintering method. The IL-GNs/PTFE flexible conductor nanocomposites were prepared by pre-stretch processing after sintering, and the tensile, thermal, and electrical properties of pre-stretched and unstretched nanocomposites were investigated. Due to the surface treatment by ionic liquid, GNs can be dispersed stably in acetone suspension for at least 24 h and uniformly distributed easily with PTFE by wet-mixing. Analyzing the results from tensile properties, Young’s modulus decreased from approximately 48 MPa for pre-stretched nanocomposites to about 18 MPa for unstretched ones. The density, shore hardness, DSC, and SEM characterizations can indicate that this variation tendency may be due to the existed microvoids in the polymer matrix and split crystalline regions after pre-stretch processing. In addition, the tensile strength of the pre-stretched nanocomposites strengthened from 14 MPa to 20 MPa in comparison with the unstretched ones, while elongation at break declined slightly from 340% to 310%. The thermal conductivity of the pre-stretched nanocomposite is slightly lower than that of the unstretched ones, which may be attributed to the transition of the nanocomposite from two-phase (polymer, fillers) before stretching to three-phase (polymer, fillers, and air) after stretching. It is also interesting to note that the pre-stretching treatment had little influence on the electrical properties of IL-GNs/PTFE nanocomposites. This may be because the treatment reduced the conductivity via destroying the conductive paths offset by the enhanced conductivity via increasing interlamellar spacing. Through simulation calculation, the pre-stretched IL-GNs/PTFE nanocomposite was shown to have a lower percolation threshold of 1.49 vol%, which achieved an approximate electrical conductivity of 5.5 × 10^−2^ s·m^−1^. The pre-stretched nanocomposite exhibits stable electrical conductivity after 180° twisting, 180° bending, and 10% stretching strain for 1000 cycles, even though it is slightly reduced by 0.30%, 0.38%, and 0.87% respectively. The volume resistivity of the nanocomposites under high temperature with stretching strain is analyzed; it is noted that the Δ*R*/*R*_0_ of the nanocomposites increases with the increase of temperature due to the PTC, and which is slightly increased for 100 stretched cycles at high temperatures. It is also observed that the growth rate of Δ*R*/*R*_0_ with low packing content is slightly higher than that with high packing content, which may be because the nanocomposites with high IL-GNs concentration have higher thermal conductivity (0.639 W·m^−1^·K^−1^ for 3.0 vol% IL-GNs versus 0.305 W·m^−1^·K^−1^ for 0.5 vol% IL-GNs). The results revealed that pre-stretch processing for IL-GNs/PTFE nanocomposite can not only significantly reduce the hardness of the material, but it can also maintain the stable electrical conductivity of the nanocomposites even after repeated deformation at ambient or high temperatures. An IL-GNs/PTFE nanocomposite flexible conductor prepared by a pre-stretching method can provide a good guideline for a wide range of applications in ambient or high-temperature strain sensors.

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
