# Peer review of "Preparation of Ionic Liquid-Coated Graphene Nanosheets/PTFE Nanocomposite for Stretchable, Flexible Conductor via a Pre-Stretch Processing"

_nanomaterials, 2019, doi:10.3390/nano10010040_

Round 1
Reviewer 1 Report
Comments;
Authors should make clear about the effect of stretching on thermal and electrical conductivity of composites? It is vague to the reader on the enhancement on electrical conductivity via pre-stretch processing with out sufficient evidences. Did authors use first time pre-stretching method to fabricate a conductor nanocomposite (IL-GNs/PTFE)? I did see enough background in the introduction part of this manuscript about the previous work regarding the pre-stretching method or any motivation to use pre-stretching technique. The x-ray diffraction peaks of IL-GNs and GNs look identical. How much differences on 2theta value? I do not believe that slightly shifting of the diffraction peak of IL-GNs towards left indicates a complete mechanical exfoliation. TEM images does not showing the correct dimension of the GNs and IL-GNs. Show the AFM images of the GNs and IL-GNs. There are differences on the unstretched and pre-stretched stress-strain curves in Fig. 4. But there is not enough effect of IL-GNs on the loading on the PTFE. Authors are suggested to make composites with pristine GNs for comparison to know the effect of modification of GNs using IL on the mechanical properties.
Author Response
Response to Reviewer 1 Comments
Point 1: Authors should make clear about the effect of stretching on thermal and electrical conductivity of composites? It is vague to the reader on the enhancement on electrical conductivity via pre-stretch processing without sufficient evidences. Did authors use first time pre-stretching method to fabricate a conductor nanocomposite (IL-GNs/PTFE)? I did see enough background in the introduction part of this manuscript about the previous work regarding the pre-stretching method or any motivation to use pre-stretching technique.
Response 1: The primary purpose of pre-stretching is to soften the PTFE substrate (We have a very deep understanding of the preparation of PTFE composites for sealing material with flexibility by pre-stretching and have published the relevant literatures, but this is really the first time that PTFE has been used as a substrate to produce a flexible conductor in our research group).Through the analysis of the experimental results, we were surprised to find that the pre-stretching treatment greatly enhanced the flexibility of the composite, while the electrical conductivity of the composite was almost constant (not improve). It is known to all that pre-stretching would break the conductive pathway in the matrix, and the conductivity of the conductor would decline, but this phenomenon did not occur in our experiment. Therefore, we hypothesized that it might be due to the interaction between the broken conductive pathway (decreased conductivity) and the increased interlamellar spacing of GNs (increased conductivity). There are relevant papers on the better conductivity of GNs with a larger interlamellar spacing in the open literature, which are cited in our manuscript. Besides, we are sure that there are little literatures on preparation of PTFE flexible conductors by pre-stretching method like us in the relevant field. This novel preparation method can not only be used to prepare PTFE flexible conductors like mentioned in our manuscript, but also can be applied in many other industrial fields that need PTFE flexible materials.
Point 2: The x-ray diffraction peaks of IL-GNs and GNs look identical. How much differences on 2 theta value? I do not believe that slightly shifting of the diffraction peak of IL-GNs towards left indicates a complete mechanical exfoliation.
Response 2: Thank you for your question. I agree with you that it is not rigorous to explain the left shift of diffraction peak by mechanical exfoliation alone. In our experiment, the primary purpose of mechanical exfoliation is to reduce the thickness of the conductive filler, while the added ionic liquid is to improve the dispersion of the filler. The movement of the diffraction peak is certainly not caused by mechanical exfoliation alone. In fact, there have also been reported that the left shift of diffraction peak is caused by the looser structure of graphite nanosheets plane (Easy preparation and characterization of graphene using liquid nitrogen and electron beam irradiation, Materials Letters, 149 (2015) 15–17, DOI: 10.1016/j.matlet.2015.02.064). In our experiment, according to the Bragg equation, the 2 Theta value is inversely proportional to the layer spacing. which means that if the value of 2 Theta decreases, the layer spacing increases. The data in Table 1 shows that both mechanical exfoliation and ionic liquids has effect on thinning GNs. For example, on the 2 Theta at around 26° (the main characteristic peaks of graphene), the value of expanded graphite (26.620) is greater than that of GNs (26.479), which indicates that mechanical exfoliation can effectively increase layer spacing of graphene nanosheets. Furthermore, the value of GNs (26.479) was greater than that of IL-GNs (26.161), which indicates that ionic liquid could also effectively increase layer spacing of graphene nanosheets. Of course, it is also impossible to completely mechanically exfoliating the expanded graphite into graphene. However, the ionic liquid acts as the interlayer inserters of GNs to weaken the interlaminar van der Waals force, while the mechanical exfoliation reduced the number of graphene nanosheet layers, both of which contributed to the left shift of the characteristic peak.
Point 3: TEM images does not showing the correct dimension of the GNs and IL-GNs. Show the AFM images of the GNs and IL-GNs.
Response 3: Thank you for the recommendation. However, during the preparation of composites, Since PTFE composites are prepared by wet-mixing, it is necessary to evenly distribute the thinner GNs in acetone solution firstly. Combined with XRD, TEM and real photos (note a. and b. in Scheme 1), we have proved that IL-GNs can be more evenly dispersed in acetone and PTFE matrix than GNs under the dual action of mechanical exfoliation and ionic liquid. Therefore, quantification of graphene dimension or layer numbers by AFM cannot better explain the dispersibility of IL-GNs in acetone. We have revised the research background in the introduction so that readers can better understand the purpose of our experiment.
Point 4: There are differences on the unstretched and pre-stretched stress-strain curves in Fig. 4. But there is not enough effect of IL-GNs on the loading on the PTFE. Authors are suggested to make composites with pristine GNs for comparison to know the effect of modification of GNs using IL on the mechanical properties.
Response 4: Thank you for the suggestion and recommendation. We have added a discussion paragraph about the effect of filler content on the tensile properties of PTFE composites: “In addition, since the maximum additive amount of filler only accounts for 3 vol% of the nanocomposite, and the low addition of filler does not affect the continuity of the PTFE matrix molecular chain. Hence, the packings have little effect on the tensile properties of nanocomposites, and which is far less than the effect of pre-stretching.”
It needs to be emphasized that: combined with our previous study, we can say with certainty that pre-stretching is the core reason for the change in tensile properties of nanocomposites, and the addition of a small amount of ionic liquid has little effect on the tensile properties of nanocomposites. The flexibility of PTFE composites is poor without pre-stretching treatment. The datum in Table 2 and Table 3 also support our conclusion. What’s more, pristine GNs cannot be mixed with PTFE powder evenly by wet-mixing method (as shown in Scheme 1 b).
However, based on reviewer’s precious recommendation, we can carry out individualized studies on the effect of ionic liquid on the mechanical properties of PTFE composites without pre-stretching.
Special thanks to you for your comments and suggestions.
Reviewer 2 Report
Although The manuscript is well written and fall on the goal and scope of the manuscript, however, some modification mentioned below is needed prior to accept.
The manuscript is mainly dealt with the ionic liquid, which was coated on graphene/PTFE composites, however, the details of ionic liquid has not been mentioned in the manuscript. Mainly function of ionic liquid and its chemical structure is essential. the author has compared the composites with stretchable and non stretchable PTFE composites, however, the thermal conductivity results didn't show significant difference. the author didn't show the difference in electrical conductivity measurement between the two composites. The author mentioned the exfoliation nature of graphene by adding ionic liquid, however XRD graph in Fig 2 do not support. Although the author has added d spacing value. lower Tensile strength has been explain by crystallinity value, DSC curve was not added to understand clearly.
Author Response
Response to Reviewer 2 Comments
Point 1: The manuscript is mainly dealt with the ionic liquid, which was coated on graphene/PTFE composites, however, the details of ionic liquid have not been mentioned in the manuscript. Mainly function of ionic liquid and its chemical structure is essential.
Response 1: First of all, thank you for your approval. Indeed, since we used wet-mixing method to prepare PTFE composites, the graphene nanosheets were first dispersed evenly in acetone solution, so ionic liquids played a crucial role in improving the dispersion ability of graphene nanosheets. And we have added the research status of ionic liquid to improve the dispersion of graphene in the introduction. The chemical structure and the promoted dispersion mechanism of ionic liquid are also discussed in the section 3.1 (such as first and last paragraph).
Point 2: The author has compared the composites with stretchable and non-stretchable, however, the thermal conductivity results didn’t show significant difference.
Response 2: Thank you for your question. It was found that the thermal conductivity of the PTFE composites was slightly reduced by pre-stretching. This may be due to the fact that the PTFE matrix has presented a phase transition (from the two phases of the matrix and the packing to the three phases of the matrix, the packing and the air) during the pre-stretching treatment, but the pre-stretching ratio is not large (just 1.5 folds) and the shrinkage appears after pre-stretching, so the introduced air phase is not much (as shown in Figure 5). Therefore, the thermal conductivity of pre-stretching composites has only slightly decreased.
Point 3: The author didn’t show the difference in electrical conductivity measurement between the two composites.
Response 3: Thank you for your question. There is no difference in the conductivity test method between the two composites. We use four-point conductivity probe with a linearly arranged four-point head on the sample surface to measure the electrical conductivity directly, no matter what kind of composite.
Point 4: The author mentioned the exfoliation nature of graphene by adding ionic liquid, however XRD graph in Fig 2 do not support. Although the author has added d spacing value
Response 4: Thank you for your question. As can be seen from Table 1, mechanical exfoliating can increase the layer spacing of expanded graphite, and the layer spacing for IL-GNs is smaller than that of GNs, which indicates that both mechanical exfoliation and ionic liquid have the effect of increasing layer spacing (The interlamellar spacing d is inversely proportional to the position of the diffraction peak, that is, the larger interlamellar spacing d is, the smaller 2 Theta value would be). Although this interpretation may be imperfect, the TEM images can also be supplementary to illustrate the effectiveness of this method. In addition, it has been reported that the left shift of diffraction peak in XRD pattern is caused by the looser structure (larger interlamellar spacing) of graphite nanosheets plane (Easy preparation and characterization of graphene using liquid nitrogen and electron beam irradiation, Materials Letters, 149 (2015) 15–17, DOI: 10.1016/j.matlet.2015.02.064).
Point 5: Lower Tensile strength has been explained by crystallinity value, DSC curve was not added to understand clearly.
Response 5: Thank you for your question and we are very sorry for our ambiguous expression, which made you confused about the tensile properties (Section 3.2). In this article, we found that the pre-stretching significantly reduced the Young’s modulus, increased the tensile strength, and slightly reduced the elongation at break of PTFE-based nanocomposite. However, by DSC analysis of crystallinity, the results indicated that the significant change in tensile properties was not caused by the change in crystallinity, but was probably mainly due to the introduction of air phase into the materials (similar to foaming). Hence, we did not place the DSC curves here that with no significant difference between all the nanocomposites, but only added the calculated crystallinity values in Table 3.
Thank you again for approving our job and precious comment.
Reviewer 3 Report
Review report on Manuscript entitled “Preparation of ionic liquid-coated graphene nanosheets/PTFE nanocomposite for stretchable, flexible conductor via a novel pre-stretch processing”
This manuscript presents interesting study on the non covalent surface modification of graphene with ionic liquid and its use to produce stretchable and flexible conductive nanocomposite. Although the presented results are interesting and novel I proposed to the authors to introduce some changes before publishing. First of all, the pre-stretching processing is one step in production of their nanocomposite and it is not novel method of production of the nanocomposites. The nanocomposites were produced by solution mixing following by cold press sintering and pre-stretching. Therefore, this should be changed accordingly this in title, abstract and manuscript text.
Here are comments to the authors:
Introduction
1.1 Line 57: The authors stated: “However, it has been reported…”
Such statement needs citation of reference where it has been reported….
1.2 The authors should add in introduction have IL been used to modify graphene and to prepare nanocomposites and what is novel in this work with respect to that.
Experimental Procedure
2.1 It is not mention if any purification step was used when IL_GNs was prepared before introducing it into nanocomposites.
2.2 In caption of Scheme 1 it is written a)…, b)… etc. In the figure there is no any a) or b) ….
Results and discussions
3.1 FTIR: Fig.1
Explain in the text what is presented in Figure 1, before you start to explain what you see in Figure1. On the other hand, in the spectra shown in Fig.1 the assignment of the peaks is already presented, so there is no need to repeat it in the text. Better to remove the assignment from the text and to add just one phrase mentioning the important peaks in IL-GNs that demonstrates presence of both IL and GNs.
In line 132-133, it is written: “…the FT-IR spectrum of GNs shows the bands at 1649 cm−1 for carbonyl (C=O), and 1552 cm−1 for phenyl (-Ph) 132 [43], which mean that GNs was mixed with graphene oxide.”
First of all, 1552 cm-1 probably refers to C=C in aromatic ring, but not phenyl. There is no phenyl rings in graphene. Secondly, presence of C=O demonstrates that graphene was partially oxidized not mixed with graphene oxide.
3.2 XRD Fig 2
The peak at 26º in all materials demonstrates the presence of stacked graphene structures, therefore it is questionable the exfoliation in GN and IL-GNs. Any comments? In such case you speak about graphite modified with IL not graphene.
Results in Table 1 are just presented, without any discussions. Add discussions in the text.
3.3 SEM images in Figure 3.
In caption of Figure 3 there is no difference between a1 and a2; and between b1 and b2. It should be explained that b1 and b2 are TEM images not SEM.
The authors started the discussion with Fig 3c. Therefore this image should be Fig 3a.
The paragraph in lines 160-166 should be rewritten, it is not clear. In Scheme 1 there is no any photo. It is difficult to follow and to understand what the message here was.
Tensile measurements: discuss why there is no any effect on the properties from the amount of added IL-GNs in the nanocomposites. Seeing the results in Fig. 4 it is not clear why the pre-stretching was selected because it makes worst the nanocmoposites. Please explain better the motivation of using pre-stretching.
SEM images of nanocomposite presented in Figure 5
First of all what is the difference between Fig 5 and Fig.7? The same figure may be used to take the separate discussions.
In Figure 5, there are no a) and b) in the figure- just in caption.
In Figure 5a, the darker part is presented as GNs, and the white one as PTFE. Usually in SEM it is oppositely. Check and correct.
There is mistake in phrase in line 220. Correct it.
According the provided explanation of the deltaR/R0 increasing after 100 stretching cycle, in lines 316-318: “…a larger thermal expansion coefficient (CTE) (especially PTFE [53]), resulting in the fracture of conductive paths 316 and increase of resistivity due to the difference in CTE between fillers and polymer matrix at a high 317 temperature”. The negative thermal expansion coefficient of graphene (Nano Lett. 2011, 11, 8, 3227-3231) is not mentioned in these explanations and certainly will have important role in the observed temperature dependent behavior.
Author Response
Response to Reviewer 3 Comments
First of all, thank you for your valuable suggestion, we have removed the word “novel” in our manuscript according to your suggestion.
Point 1.1: Line 57: The authors stated: “However, it has been reported…”. Such statement needs citation of reference where it has been reported…
Response 1.1: Thanks to you for your comment. It is really not rigorous not to quote here. We have quoted the relevant literatures at the end of this sentence.
Point 1.2: The authors should add in introduction have IL been used to modify graphene and to prepare nanocomposites and what is novel in this work with respect to that.
Response 1.2: Thank you for the recommendation. We have added the research background of ionic liquid modified graphene and the experimental objective of this chosen modification method in the introduction as you suggested.
Point 2.1: It is not mention if any purification step was used when IL-GNs was prepared before introducing it into nanocomposites.
Response 2.1: Thank you for your question. In this experiment, IL-GNs was not purified before mixing PTFE powder with IL-GNs in acetone solution by wet mixing method. IL-GNs powders are obtained by directly drying the mechanically exfoliated IL / alcohol / GNs emulsion at 70 °C for 2 hours, and it has been written in the text
Point 2.2: In caption of Scheme 1 it is written a)…, b)… etc. In the figure there is no any a) or b) ….
Response 2.2: Thank you for your question. Note a) and b) are the text with a green background in Scheme 1.
Point 3.1: Explain in the text what is presented in Figure 1, before you start to explain what you see in Figure 1. On the other hand, in the spectra shown in Fig.1 the assignment of the peaks is already presented, so there is no need to repeat it in the text. Better to remove the assignment from the text and to add just one phrase mentioning the important peaks in IL-GNs that demonstrates presence of both IL and GNs.
In line 132-133, it is written: ‘…the FT-IR spectrum of GNs shows the bands at 1649 cm-1 for carbonyl (C=O), and 1552 cm-1 for phenyl (-Ph), which mean that GNs was mixed with graphene oxide.’
First of all, 1552 cm-1 probably refers to C=C in aromatic ring, but not phenyl. There is no phenyl rings in graphene. Secondly, presence of C=O demonstrates that graphene was partially oxidized not mixed with graphene oxide.
Response 3.1: Thank you for your correction and instruction. We have rewritten this paragraph according to your suggestion.
Point 3.2: The peak at 26° in all materials demonstrates the presence of stacked graphene structures, therefore it is questionable the exfoliation in GN and IL-GNs. Any comments? In such case you speak about graphite modified with IL not graphene.
Results in Table 1 are just presented, without any discussions. Add discussions in the text.
Response 3.2: Thank you for your question. Whether graphene, expanded graphite or graphite, characteristic peaks appear at about 26° on the XRD pattern. It has been reported that the looser structure (larger interlamellar spacing) of the graphite nanosheets plane, the diffraction peak moves to the left (Easy preparation and characterization of graphene using liquid nitrogen and electron beam irradiation, Materials Letters, 149 (2015) 15–17, DOI: 10.1016/j.matlet.2015.02.064). And we have supplemented discussion paragraph for the calculation results form Table 1 according to your suggestion.
Point 3.3-1: In caption of Figure 3 there is no difference between a1 and a2; and between b1 and b2. It should be explained that b1 and b2 are TEM images not SEM.
The authors started the discussion with Fig 3c. Therefore, this image should be Fig 3a.
Response 3.3-1: Thank you for your question. We have changed the graphic label of expanded graphite from 3c to 3a, and explained the b1 and b2 belongs to IL-GNs, and the c1 and c2 belongs to GNs. In addition, we have explained in the caption that b1 and c1 are SEM, while b2 and c2 are TEM.
Point 3.3-2: The paragraph in lines 160-166 should be rewritten, it is not clear. In Scheme 1 there is no any photo. It is difficult to follow and to understand what the message here was.
Response 3.3-2: Thank you for your question. This paragraph is a supplement to SEM and TEM with real photos, indicating that ionic liquid and mechanical exfoliation have successful contribution to the dispersion of GNs in acetone and PTFE powders. The real photos a) and b) are shown in text with a green background in Scheme 1.
Point 3.3-3: Tensile measurements: discuss why there is no any effect on the properties from the amount of added IL-GNs in the nanocomposites. Seeing the results in Fig. 4 it is not clear why the pre-stretching was selected because it makes worst the nanocomposites. Please explain better the motivation of using pre-stretching.
Response 3.3-3: Thank you for your question. We have added a discussion on why the amount of packing has little effect on the tensile properties of composites according to your suggestion: “In addition, since the maximum additive amount of filler only accounts for 3 vol% of the nanocomposite, and the low addition of filler does not affect the continuity of the PTFE matrix molecular chain. Hence, the packings have little effect on the tensile properties of nanocomposites, and which is far less than the effect of pre-stretching.”
Pre-stretching is a key step in our experiment. And the purpose of pre-stretching as we mentioned in the introduction: “In addition, due to the very regular molecular chains of PTFE and high crystallinity after sintering, the PTFE matrix composite has a relatively high hardness that could not be suitable as flexible material. However, through our previous study, it was found that the PTFE-based composites were stretched within a specific range for a fast speed at ambient or high temperature, the characteristics of material would change from rigidity to flexibility.” Our research group has published related literatures on the softening of PTFE by pre-stretching treatment. Without pre-stretching treatment, the flexibility of PTFE composite is not good. In addition, pre-stretching only slightly reduces the elongation at break of the material by 8.8%, which would hardly affect the ductility of the composite at a no huge tensile ratio as a flexible conductor. In conclusion, the advantages of pre-stretching process far outweigh the disadvantages for PTFE flexible conductors.
Point 3.3-4: SEM images of nanocomposite presented in Figure 5. First of all, what is the difference between Fig 5 and Fig.7? The same figure may be used to take the separate discussions. In Figure 5, there are no a) and b) in the figure- just in caption. In Figure 5a, the darker part is presented as GNs, and the white one as PTFE. Usually in SEM it is oppositely. Check and correct.
Response 3.3-4: Thank you for your question. Although Figure 5 and Figure 7 are similar, they are actually different and have different meanings. Figure 5 is intended to show that the PTFE matrix would have microporous structure after pre-stretching, that is a proof of matrix softening. It can be seen from figure 5 that the matrix of the unstretched composite is compact, while the stretched matrix is slightly fluffy. Figure 7 is intended to illustrate that pre-stretching also increases the distance between IL-GNs, resulting in the disconnection of the conductive or thermal pathways. Figure 7 shows that the packings in the unstretched composite is relatively compact and the separation appears obviously after pre-stretching. We have revised the caption in Figure 5 and Figure 7 as your recommendation, that could help readers to understand easily. As to which part in Figure 5a is IL-GNs and which part is PTFE, we can definitively say that the darker part represents IL-GNs and the white one represents PTFE, because the white one can be shown typical PTFE micropores after pre-stretching.
Point 3.3-5: There is mistake in phrase in line 220. Correct it.
Response 3.3-5: Thanks for your reminding. We have revised the phrase according to your reminder.
Point 3.3-6: According the provided explanation of the delta R/R0 increasing after 100 stretching cycle, in lines 316-318: “…a larger thermal expansion coefficient (CTE) (especially PTFE), resulting in the fracture of conductive paths and increase of resistivity due to the difference in CTE between fillers and polymer matrix at a high temperature”. The negative thermal expansion coefficient of graphene (Nano Lett. 2011, 11, 8, 3227-3231) is not mentioned in these explanations and certainly will have important role in the observed temperature dependent behaviour.
Response 3.3-6: Thanks for your reminding and suggestion. Your suggestion is very valuable. In our initial discussion of the results, we have considered the negative thermal expansion coefficient of graphene effect on the electrical properties of composite, but due to the thermal expansion coefficient of pure PTFE theoretical value (approximate 100 × 10-6 K-1) is 12.5 times that of the graphene (approximate -8 × 10-6 K-1), and the PTFE matrix is transformed into a microporous structure after the pre-stretching, therefore, the thermal expansion coefficient of the PTFE matrix with large volume content in stretched composites should be greater than the theoretical value. Hence, for our experiment, the effect of PTFE matrix on the reduced conductivity should be far greater than that of fillers. In addition, different layers of graphene have different thermal expansion coefficients, so if we were to discuss the negative thermal expansion coefficient of graphene with unknown layer numbers, the results would be very difficult to predict. We think it’s actually possible to do a special study on the thermal expansion coefficient of unstretched PTFE / graphene composites.
However, we have revised some sentences to make the paper more informative and rigorous.
Thank you again for your valuable advice.
Round 2
Reviewer 2 Report
Thanks for improving according to the suggestion.
I still believe that crystallinity of the composites has great influence on mechanical properties. It will be great if you can corelate the mechanical strength (TS) with your crystallinity results.
Author Response
Thank you for your question. In previous studies, we conducted special studies on the tensile properties of PTFE/short glass fibers composites (packings with 15% mass fraction), and found that different tensile ratios and different tensile rates would affect the crystallinity and tensile strength of the composites. The results indicated that, with the increase of the tensile rate, the crystallinity increases while tensile strength increases first and then decreases (The tensile strength of all stretched PTFE/short glass fibers composites is greater than that of the unstretched ones). With the increase of tensile ratio, the crystallinity decreases first and then increases while tensile strength increases. Although we did not specifically discuss the relationship between crystallinity and tensile strength in previous studies, it can be concluded that pre-stretching treatment would affect the crystallinity and enhance the tensile strength of PTFE composites. Consistent with the results of previous manuscripts, the increase of the tensile strength in this manuscript is due to the pre-stretching effect which contributes to the orientation of PTFE matrix molecular chains. However, by using the same pre-stretching method, the crystallinity of the stretched composite hardly changed compared with that of the unstretched one. The change trend of crystallinity in this experiment is different from that in previous studies. We speculated that this may be due to two reasons: First, compared with graphene, the short glass fiber as an inorganic filler has better heterogeneous nucleation in PTFE matrix, and pre-stretching make packings debonding from matrix, and the crystal zone around the packing caused by heterogeneous nucleation has a less destruction during debonding, so although a large number of big cavities due to packings debonding to soften PTFE composites greatly, the crystallinity increase in our previous studies. Second, in this experiment, due to the low content of graphene in the composite (maximum volume fraction at 3%), the phenomenon of graphene debonding in composite after pre-stretching is not significant, so the softening of the composite may mainly depend on the tiny cavities in PTFE matrix after the pre-stretched (rather than the big cavities caused by packing debonding). Therefore, in conclusion, we speculate that the softening of the material is not caused by the change in crystallinity in this experiment. In addition, although pre-stretching can destroy crystal regions in principle, the pre-stretching treatment in this experiment was carried out in a high temperature environment, where the molecular chains have higher activity and are easier to rearrange after being stretched, and the rearrangement of the molecular chains is conducive to recrystallization. In summary, pre-stretching treatment has no significant effect on crystallinity of graphene/PTFE nanocomposites, which is really an interesting phenomenon.
However, we do appreciate your advice, which is very valuable. We have added the discussions for the increased tensile strength and the decreased elongation at break in the section of tensile properties discussion. Besides, we will conduct an in-depth special investigate on the relationship between crystallinity and mechanical strength (TS) of stretched PTFE-based composites in the future research.
Reviewer 3 Report
Authors have thoroughly corrected the manuscript as it was adviced and I propose publishing as it is.
Author Response
Thank you for your support for our article. I have revised the language of the article.